# Clinicopathological Profiles and Survival Outcomes of Patients with Gastric Cancer According to the Borrmann Endoscopic Classification: A Single-Center Retrospective Cohort Study

**DOI:** 10.3390/medicina61112032

**Published:** 2025-11-14

**Authors:** Andrés Camilo Pachón-Mendoza, Oscar Daniel Pacheco-Can, Felipe Angulo-Várguez, Dayana Williams-Jacquez, Marlene Chaurand-Lara, Ana Ligia Gutiérrez-Solis, Azalia Avila-Nava, Mariana Irigoyen-Anguiano, Rodolfo Chim-Aké, Katy Sánchez-Pozos, Roberto Lugo

**Affiliations:** 1Unidad de Investigación, Hospital Regional de Alta Especialidad de la Península de Yucatán, IMSS-Bienestar, Mérida 97130, Mexico; andres074@outlook.com (A.C.P.-M.); oscar_dpc@outlook.com (O.D.P.-C.); ganaligia@gmail.com (A.L.G.-S.); zomi33@gmail.com (A.A.-N.); rodolfochim@hotmail.com (R.C.-A.); 2Departamento de Hematología, Centro Médico Nacional 20 de Noviembre, Instituto de Seguridad y Servicios Sociales de los Trabajadores del Estado (ISSSTE), Ciudad de México 03104, Mexico; 3Becario de la Dirección General de Calidad y Educación en Salud (DGCES), Secretaría de Salud, Ciudad de México 11400, Mexico; 4Departamento de Gastroenterología, Hospital Regional de Alta Especialidad de la Península de Yucatán, IMSS-Bienestar, Mérida 97130, Mexico; favidoctor@hotmail.com; 5Departamento de Anatomía Patológica, Hospital Regional de Alta Especialidad de la Península de Yucatán, IMSS-Bienestar, Mérida 97130, Mexico; williamsdayana@gmail.com; 6Servicio de Endoscopía Digestiva, Hospital Regional de Alta Especialidad de la Península de Yucatán, IMSS-Bienestar, Mérida 97130, Mexico; dra.chaurand@gmail.com; 7Serología Banco de Sangre, Hospital General de Zona 1 del Instituto Mexicano de Seguro Social (IMSS), Villa de Álvarez, Colima 28984, Mexico; mirigoyenanguiano@gmail.com; 8División de Investigación, Hospital Juárez de México, Ciudad de México 07760, Mexico; katypozos@gmail.com

**Keywords:** Borrmann classification, endoscopic diagnosis, gastric cancer, survival, Mexican population

## Abstract

*Background and Objective*: Gastric cancer (GC) is a serious public health problem in southeastern Mexico. Some cases go undiagnosed or are diagnosed at advanced stages of the tumors. Borrmann classification is the method used by endoscopists to classify gastric lesions and identify tumor stage. This study aimed to characterize GC patients treated at a specialized hospital in the Yucatan Peninsula, Mexico, according to the Borrmann endoscopic classification, with a focus on clinicopathological characteristics and survival differences. *Materials and Methods*: A retrospective cohort study was conducted among patients aged 18 years or older who underwent an endoscopic procedure at the hospital to confirm a diagnosis of GC between January 2019 and December 2024. Clinical data were collected, including medical history, blood type, non-communicable diseases, tumor type, tumor location (primary or metastatic), and details of medical and/or surgical treatment. Survival curves were generated for all patients and stratified by the Borrmann classification. *Results*: A total of 209 cases of GC were included, with 115 men with a mean age of 59.3 years and 94 women with a mean age of 52.2 years. Acid peptic disease (70.3%), followed by wasting syndrome (66.9%), was the most common medical condition in patients with GC. Blood type O with a positive Rh factor was the most frequent (66.5%). According to the Borrmann classification, localized tumors (*p* = 0.001) were observed at lower Borrmann levels, whereas *Helicobacter pylori* (*p* = 0.040) was more frequent at higher levels. The overall survival time was 18 months for all patients; specifically, 18 months at higher Borrmann levels and 20 months at lower levels. *Conclusions*: GC is a highly prevalent malignancy in southeastern Mexico. The Borrmann classification remains a valuable and practical tool for evaluating GC. The association between Borrmann endoscopic classification and the clinicopathological and survival characteristics may contribute to accurate diagnosis assessment and improved prognostic stratification in future GC cases.

## 1. Introduction

According to the latest GLOBOCAN 2022 report, gastric cancer (GC) ranks sixth in incidence and seventh in mortality worldwide [1,2]. GC remains a public health problem in several regions of the world, including Latin America. Due to its epidemiological and molecular characteristics, GC in this population has not yet been fully characterized [3]. The process of carcinogenesis is a multistep event that begins with atrophic gastritis, progresses to premalignant lesions, and ultimately leads to neoplasia [4,5]. These events involve several risk factors, among which *Helicobacter pylori* (*H. pylori*) infection is considered the primary one, followed by dietary habits, genetic susceptibility, obesity, gastroesophageal reflux disease, tobacco and alcohol consumption, and population ethnicity [2,5]. GC diagnosis is performed by histopathological evaluation of endoscopic biopsy specimens. The Borrmann classification is the most widely used method among endoscopists to classify gastric lesions based on macroscopic characteristics. Among these, Borrmann types III and IV are strongly associated with large tumor size, serosal invasion, and poor overall survival compared with types I and II [5,6,7].

In countries such as South Korea, China, and Japan, where GC incidence is high, endoscopic screening has been established as a routine test over the past two decades to reduce the incidence of advanced tumor stages and mortality rates [8,9,10]. In contrast, in the United States of America and Latin America, screening programs for GC remain very limited, possibly due to their high costs and uncertain effectiveness in these populations. Consequently, implementing large-scale screening programs is challenging. Venezuela and Chile are the only Latin American countries with programs for detecting *H. pylori* antigens through serological screening. However, the effectiveness of this type of detection has not been fully determined [11,12]. Moreover, in Mexico, there is still no national record of GC incidence and mortality. Specifically, the population of southeastern Mexico faces several risk factors, including comorbidities such as diabetes and obesity, as well as an ancestral diet characterized by high consumption of foods rich in spices, salt, and chili [13]. Genetic factors may also contribute to disease susceptibility. However, there is a lack of scientific evidence describing and characterizing this neoplasm, particularly regarding its sociodemographic and molecular aspects [14]. In addition, a large proportion of the population remains undiagnosed or is diagnosed at late stages of the disease, which limits treatment options and increases mortality. Recent reports have identified the southeastern Mexico region as a high-incidence region compared with the national average (6.4 vs. 3.3 cases per 100,000 inhabitants) [15,16]. Therefore, the objective of this study was to characterize GC patients treated at a specialized hospital in the Yucatan Peninsula, Mexico, according to the Borrmann endoscopic classification, with emphasis on the clinicopathological characteristics and survival differences.

## 2. Materials and Methods

A retrospective cohort and observational study was conducted among patients over 18 years of age diagnosed with GC and treated at a specialized hospital in Southeastern Mexico between January 2019 and December 2024.

### 2.1. Selection of the Study Participants

Clinical information, including diagnosis, hospitalization, and discharge, was systematically collected for all participants. Additionally, telephonic follow-up and monitoring of subsequent medical appointments were performed to assess outcomes.

Patients were included if they were diagnosed with a primary GC tumor during hospitalization, underwent upper gastrointestinal endoscopy to confirm the diagnosis, and had complete clinical records. All endoscopic procedures were performed by certified gastroenterologists using standard video endoscopes. The stomach was systematically inspected, and multiple biopsies were obtained from lesions for histopathological confirmation. Tumors were classified according to the Borrmann classification based on endoscopic and histopathological findings, categorizing lesions as type I (polypoid), type II (ulcerated with defined margins), type III (ulcerated with infiltration), or type IV (diffusely infiltrating).

Patient data, including age, place of birth, occupation, and lifestyle, were recollected during hospitalization and at the time of tumor diagnosis to characterize the study population. Additional information on comorbidities (hypertension, diabetes mellitus, and dyslipidemias), blood type, and body mass index (BMI) was obtained from clinical records.

Data on tumor type and location (primary and metastatic), as well as details of medical (chemotherapy and/or radiotherapy) and surgical treatments (partial or total gastrectomy), and subsequent adjuvant treatments (chemotherapy, radiotherapy, or palliative treatments) administered during hospitalization or after discharge were also collected.

Patients were excluded if they lacked information on *H. pylori* status, underwent endoscopies at other hospitals, had no confirmed endoscopic diagnosis of GC, or had incomplete clinical data regarding medical, surgical, or adjuvant treatments. Those diagnosed with malignant or benign tumors different than GC were also excluded from the study.

Survival was defined as patients who were discharged after hospital treatment and could be contacted to confirm subsequent medical visits and follow-up medical treatments (chemotherapy/radiotherapy). Mortality included patients who died during hospitalization due to GC, those discharged due to no further therapeutic benefit, and those who left the hospital against medical advice and subsequently died. Survival curves were generated only for patients who met the survival follow-up criteria. The time elapsed (in months) from hospital discharge to death by GC was recorded and verified in the medical records. In addition, the time to the last medical visit, the last treatment received (in cases where the patient discontinued treatment), or death due to other causes (censored event/lost to follow-up) was also documented.

### 2.2. Statistical Analysis

Descriptive statistical analysis was performed using Jamovi v2.3 and RStudio v2025.05.1. Sociodemographic and clinical variables, as well as comparisons of Borrmann classifications, were analyzed using the Chi-square test. Student’s *t*-test was used to examine age. Survival analyses were conducted using the Log-Rank test, and multivariable survival analysis was performed using Cox proportional hazards regression. The *p*-values < 0.05 were considered statistically significant.

## 3. Results

A total of 209 patients with GC who met the inclusion criteria were included, comprising 94 women (44.9%) and 115 men (55.1%) (Figure 1). The mean age of the participants was 56.0 ± 14.7 years. Most patients originated from the states of the Yucatan Peninsula (Yucatan, Campeche, and Quintana Roo), while only 2.8% came from other regions of the country (Table 1). Among the toxicological habits reported, regular alcohol consumption was the most frequent, whereas marijuana use was the least common, reported in 11.4% of patients.

Occupational data from this study showed that 48.3% of participants held unskilled jobs, a pattern more common among males than females. In addition, a similar proportion of patients (46.4%) reported being unemployed. Interestingly, the majority of this group were female patients, many of whom assume household responsibilities. These findings are consistent with the profile of patients attending this study, as the hospital is the only specialized medical center in southeastern Mexico that provides care to uninsured patients from the three states of the Yucatan Peninsula.

Only 35.8% of patients arrived at the hospital with a prior diagnosis of GC, and 40.6% had undergone a previous endoscopic examination; the remaining participants were diagnosed during hospitalization. The most common medical conditions observed before admission were acid peptic disease (63.2%), wasting syndrome (22.0%), and upper gastrointestinal bleeding (11.0%). These proportions largely remained unchanged at the time of hospitalization, except for an increase in the prevalence of wasting syndrome (Table 2). During clinical evaluation, 71.2% of patients reported regular use of proton pump inhibitors, 2.3% had a family history of GC, and 44.4% had undergone axial tomography (thorax, abdomen, and pelvis) before hospital admission.

Over 40% of patients presented with two or more medical conditions during hospitalization. Common combinations included acid peptic disease with wasting syndrome, wasting syndrome with bleeding of the digestive tract, or acid peptic disease with dysphagia. Abdominal pain was reported in approximately 60% of patients, but it was not classified as a principal medical condition due to its nonspecific nature (Table 2). In all cases, the patient’s primary symptom corresponded to the principal medical condition.

Chronic-degenerative diseases were infrequent among GC patients, with low prevalence of diabetes, hypertension, and dyslipidemia. Only 25.8% of patients were overweight, and 7.6% obese. Blood type O was the most prevalent, followed by A positive.

According to Borrmann’s endoscopic classification, 51.2% of advanced tumors (Borrmann III–IV) were diagnosed for the first time during hospitalization. Patients with advanced Borrmann levels showed higher proportions of tobacco, alcohol, and marijuana consumption, as well as acid peptic disease, wasting syndrome, and upper gastrointestinal bleeding (Table 3). Localized tumors were more frequent in lower Borrmann levels (I–II), while metastatic tumors predominated in advanced levels (III–IV).

Although Table 3 shows the type of treatment received during hospitalization, most patients who underwent surgical treatment also received complementary or adjuvant therapies. Among patients with low Borrmann classification, only 29.4% (5/17) received surgical treatment (partial or total gastrectomies) as the primary therapy; 23.5% (4/17) received adjuvant chemotherapy, and 47.1% (8/17) received both chemotherapy and radiotherapy as adjuvant treatment. In contrast, among patients with high Borrmann classification, 36.9% (38/103) underwent surgery alone; 43.6% (45/103) received adjuvant chemotherapy; and 19.4% (20/103) received both chemotherapy and radiotherapy as adjuvant therapy.

Primary tumors were most often located in the body of the stomach, followed by the esophagogastric junction and the antrum. The main metastatic sites were the lymph nodes, peritoneum, and liver (Figure 2).

Only 50 patients (23.9%) met the criteria for survival, with a median overall survival of 18 months (Figure 3). When stratified by Borrmann classification, median survival was approximately 20 months for types I–II, and 18 months for types III–IV, with no statistically significant difference by log-rank test (*p* = 0.373) (Figure 4). Among survivors, 48% had a previous GC diagnosis, 36% were Borrmann I–II, 88% had localized tumors, and 82% underwent surgical resection (total or partial gastrectomy) as corrective treatment.

Conversely, 76.1% (159/209) of the patients died: 35.7% (10/28) with low Borrmann classification, and 82.3% (149/181) with high Borrmann classification. As mentioned above, some patients received adjuvant treatments during hospitalization. In this context, 90% (9/10) of patients who died with low Borrmann classification had received palliative treatment, while 81.9% (122/149) of those who died with high Borrmann classification received palliative care.

To determine survival according to the Borrmann classification in patients with GC, Cox regression analyses were performed. However, no association was found between survival and low Borrmann (*n* = 18; events = 11; censored = 7) vs. high Borrmann (*n* = 32; events = 26; censored = 6) classification (HR = 0.58; 95% CI [0.26–1.30], *p* = 0.188). After adjusting for age, sex, and treatment type (surgical vs. medical), the association remained non-significant (HR = 0.46; 95% CI [0.20–1.06], *p* = 0.068). Finally, when additionally adjusting for comorbidities (hypertension, diabetes mellitus, dyslipidemia, and obesity), no association was observed (HR = 1.54; 95% CI [0.52–4.55], *p* = 0.435).

## 4. Discussion

This study provides the first description of the sociodemographic and epidemiological characteristics of GC in the population of Southeastern Mexico. The study was conducted at a specialized hospital in southeastern Mexico that is part of the Mexican government’s health services. The hospital’s area of influence covers the states of Yucatán, Campeche, and Quintana Roo, with a total population of more than 3 million inhabitants, making it a referral hospital in the region [17]. The population predominantly lives in rural areas, has low socioeconomic status, and has limited educational attainment. These factors likely contribute to inadequate self-care of health and late-stage disease diagnosis, as reflected by the high proportion (64.2%) of cases diagnosed after clinical presentation. Moreover, improvements are needed in hospital clinical practices and in the completeness and accuracy of clinical reports.

A total of 209 cases met the inclusion criteria, with a predominance of males and a mean age of 56 years. Most participants resided in the states of the Yucatan Peninsula. Only 35.8% had a prior diagnosis of GC, while 40.6% had undergone previous endoscopic examination. Most of the males were employed in unskilled occupations and were more frequently exposed to risk factors, including construction dust, microparticles, and pesticides (41.6%). In this regard, previous studies have reported that occupations such as mining, agriculture, and construction are associated with an increased risk of mortality from GC [18,19]. In contrast, female patients with unskilled occupations (6.7%) and unemployed (35.9%) were more often exposed to concentrated cleaning agents. These occupational factors may contribute to the development of GC. However, in the present study, it was not possible to determine a direct relationship between the degree of the agent’s exposure and the risk of tumor development, as these characteristics were not explicitly inquired about prior to hospitalization.

On the other hand, high levels of alcohol consumption, along with elevated tobacco and marijuana use, were observed. These findings are consistent with previous reports, as exposure to these substances produces additive cytotoxic effects that contribute to cellular DNA damage and the development of premalignant lesions [4,15,20]. However, the degree of exposure to these toxins was not quantified during the clinical evaluation, which limits the ability to determine consumption levels or to correlate them with tumor grade.

Patients with GC are typically asymptomatic in the early stages; symptoms usually appear once the disease has progressed to advanced stages [21]. Many patients may mistake GC symptoms for abdominal pain or chronic gastritis. Moreover, widespread use of proton pump inhibitors (PPIs) to alleviate these symptoms may inadvertently increase GC risk. Recent studies suggest that chronic PPI use is associated with GC development, primarily because reduced gastric acidity facilitates bacterial colonization and increases nitrosamine production, which are recognized risk factors for gastric adenocarcinoma. Additionally, hypochlorhydria induces hypergastrinemia, stimulating the proliferation of enterochromaffin-like cells, thereby increasing the risk of atrophic gastritis and the formation of gastric polyps [22,23,24]. Among GC patients, acid peptic disease was the most frequent comorbidity at hospital admission and during hospitalization, indicating that the early acid-related symptoms did not prompt clinical evaluation. Instead, progressive signs such as weight loss (the second most common clinical feature) and gastrointestinal bleeding eventually led patients to seek care. Dysphagia was also observed, typically associated with tumors at the gastroesophageal junction [25]; however, only about one-fifth of the patients presented with this symptom.

The prevalence of non-communicable diseases (NCDs), including systemic arterial hypertension, type 2 diabetes, and dyslipidemia, was 14.8%, lower than reported in Southeastern Mexico [26,27,28]. This may reflect negative metabolic balance from tumor metabolism and reduced oral intake, leading to decreased BMI and partial self-compensation of these conditions [29]. Nevertheless, nearly 33% of patients were overweight or obese, despite many presenting with wasting syndrome.

Blood type has long been associated with GC risk [30]. Blood type A is linked to a 30–40% higher risk of intestinal metaplasia or gastric dysplasia [31,32], while type O may indirectly increase risk through susceptibility to peptic ulcer disease [30]. In our cohort, blood type O with a positive Rh factor was most prevalent, followed by type A, consistent with the general Mexican population [15]. Anemia, resulting from chronic gastrointestinal blood loss, was observed in just over 40% of patients and represents a key alarm sign of gastrointestinal malignancy [33]

Endoscopy is the primary method for diagnosing GC in patients with suspected symptoms, enabling early detection and potential curative treatment, as demonstrated in Asian populations with systematic screening [34]. The Borrmann classification assesses the macroscopic grade of gastric lesions, with low levels associated with localized tumors and better prognosis, and high levels indicating advanced or metastatic disease often requiring surgery. High Borrmann levels are further associated with larger tumor size, greater invasion of adjacent organs, and reduced response to medical treatments, including chemotherapy, leading to poorer outcomes [7,15,35]. Initially designed to standardize endoscopic findings, the classification is increasingly used as a prognostic tool, particularly in resource-limited settings [7,36].

*H. pylori* infection is a recognized risk factor for GC. In our study, a relatively low prevalence of *H. pylori* was observed, likely due to reliance on histopathological examination of endoscopic tissue samples. Because the bacterium exhibits centripetal colonization, specimens that did not include the lesion’s peripheral margins may have produced false-negative results [37,38]. Interestingly, *H. pylori* infection was associated with high Borrmann levels, suggesting a greater likelihood of metastatic disease, potentially reflecting population-specific factors such as limited access to timely diagnostics or suboptimal antimicrobial therapy. However, our findings contrast with other studies, which report that untreated *H. pylori* infection induces gastric mucosal inflammation, promotes carcinogenesis, and increases tumor aggressiveness. Eradication therapy reduces, but does not eliminate, the risk of primary or metachronous GC [37,39].

Regarding metastatic patterns, the most commonly affected sites in the present study were, in descending order, lymph nodes, peritoneum, liver, lungs, intestine, and, less frequently (<5%), bone, bile ducts, and adrenal glands [40]. Tumor location influences tropism, with esophagogastric junction tumors preferentially spreading to the liver, pyloric tumors to the lungs, and antral tumors occasionally to the bone.

Despite significant progress in elucidating tumor biology and in the development of minimally invasive therapies, surgical resection remains the gold standard for curative treatment, typically via radical or partial gastrectomy [41,42]. Only a limited number of patients are suitable for curative-intent surgery due to localized disease. In advanced stages, treatment is personalized, often combining chemotherapy and surgery, while palliative care remains the only option in severe cases. Surgical intervention improves survival in localized disease but does not confer benefit in metastatic or advanced-stage GC [43].

Overall survival in this study was 18 months, with minimal differences between patients with high (18 months) and low (20 months) Borrmann classifications. These outcomes likely reflect late-stage diagnosis, high prevalence of comorbidities, delayed access to healthcare, and lifestyle and dietary factors. Although no association was observed between survival and the Borrmann endoscopic classification (low vs. high), these findings represent the first reported survival estimates for this population and are consistent with prior Mexican studies, which have documented survival ranging from 7 months in metastatic disease to 32 months in locally advanced disease, and up to 39 months for localized tumors [44]. The limited differences observed may be attributed to the sample size, the advanced stage of disease at diagnosis, tumor severity, and treatment discontinuation (chemotherapy and/or radiotherapy) among some patients.

Key study limitations included incomplete medical records for a substantial proportion of patients and pre-diagnostic mortality due to delayed symptom recognition or restricted access to healthcare services. These findings highlight the urgent need for mandatory endoscopic screening programs for individuals aged 50 or older with risk factors or early gastrointestinal symptoms, a strategy successfully implemented in several Asian countries that has led to reduced GC incidence.

## 5. Conclusions

This study confirms that GC is highly prevalent in Southeastern Mexico. Acid peptic disease was the most common symptom observed before and during the hospitalization. Blood type O with positive Rh factor was the most frequent in this population. More GC cases were diagnosed at advanced stages of the disease, showing an overall survival of 18 months. The Borrmann endoscopic classification correlated with the extent of disease at presentation (metastatic involvement was more frequent in types III–IV), but in this cohort, it was not independently associated with survival after adjustment for age, sex, and treatment. However, the Borrmann classification remains the most widely used method for classifying gastric lesions based on macroscopic characteristics. The study supports the need to implement early detection strategies and earlier access to curative treatment.

## Figures and Tables

**Figure 1 medicina-61-02032-f001:**
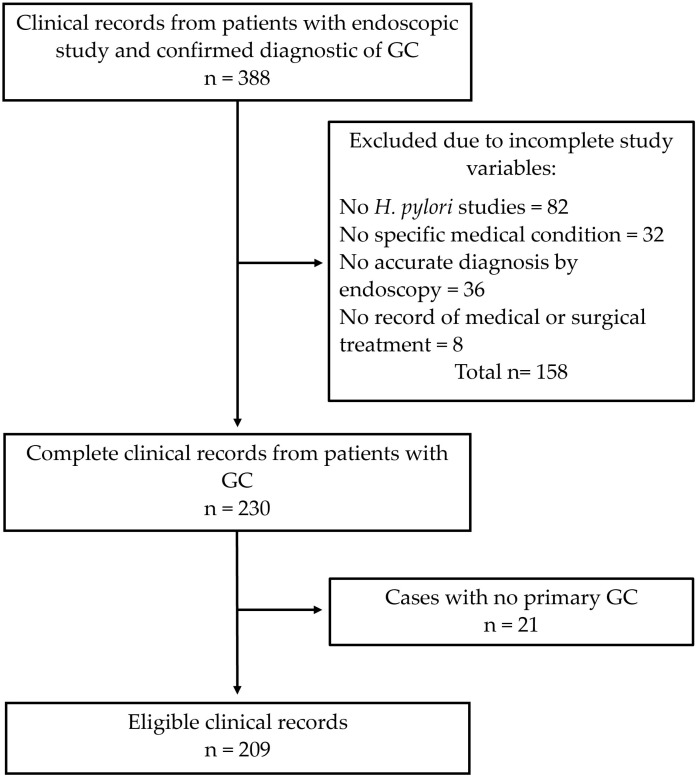
A schematic representation of the sample selection. GC: gastric cancer.

**Figure 2 medicina-61-02032-f002:**
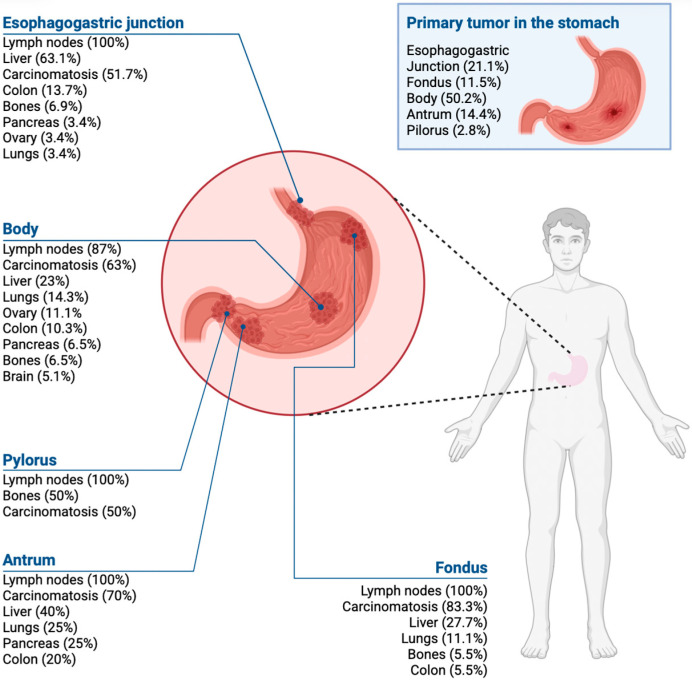
Gastric cancer metastases according to the tumor location in the stomach.

**Figure 3 medicina-61-02032-f003:**
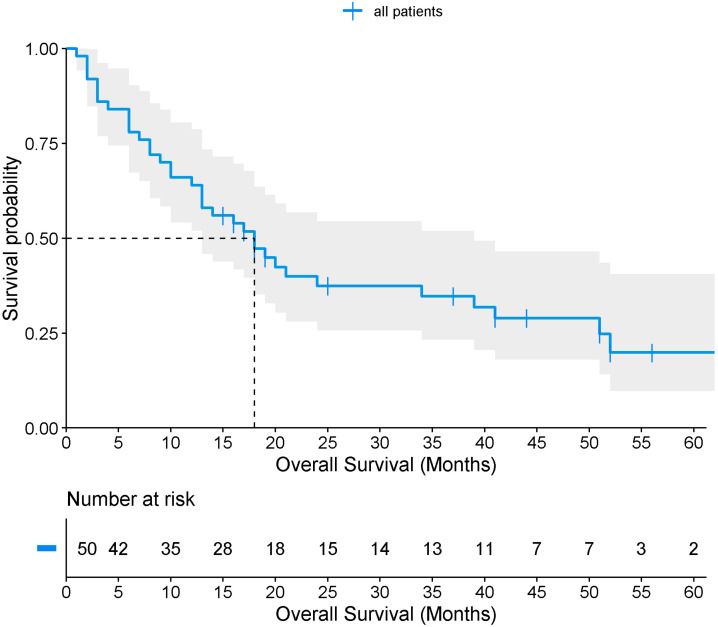
Overall survival of patients with gastric cancer.

**Figure 4 medicina-61-02032-f004:**
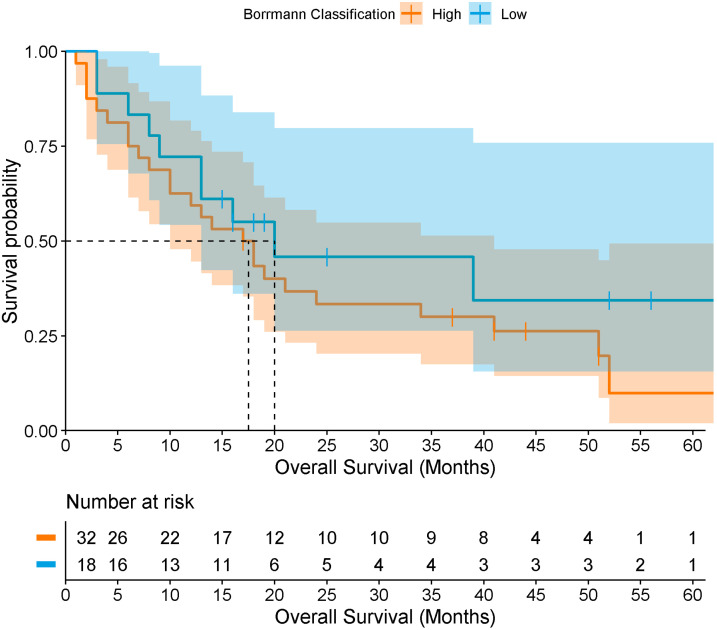
Overall survival of patients with gastric cancer according to low and high Borrmann classification levels. The darker region reflects overlap between the confidence bands of both groups.

**Table 1 medicina-61-02032-t001:** The sociodemographic characteristics of the population.

Characteristics	Total *n* = 209	Men *n* = 115	Women *n* = 94	*p*-Value
Age in years (mean ± SD)	56.1 ± 14.7	59.3 ± 14.7	52.2 ± 13.8	<0.001
Age (minimum/maximum)	19–87	22–87	19–87	
Place of birth (States of Mexico), *n* (%)				
Yucatan	160 (76.6)	90 (43.1)	70 (33.5)	0.520
Campeche	17 (8.2)	10 (4.8)	7 (3.4)	0.742
Quintana Roo	26 (12.4)	13 (6.2)	13 (6.2)	0.582
Others ^†^	6 (2.8)	2 (0.9)	4 (1.9)	0.279
Occupation, *n* (%)				
Employed with a skilled job	11 (5.3)	6 (2.9)	5 (2.4)	0.512
Employed with an unskilled job ^‡^	101 (48.3)	87 (41.6)	14 (6.7)	<0.001
Unemployed	97 (46.4)	22 (10.5)	75 (35.9)	<0.001
Lifestyle (substance used behaviors), *n* (%)				
Tobacco uses	76 (36.0)	67 (31.7)	9 (4.3)	<0.001
Alcohol consumption	106 (50.2)	74 (35.0)	32 (15.2)	<0.001
Marijuana consumption	24 (11.4)	23 (10.9)	1 (0.5)	<0.001

^†^ Other states of Mexico: Chiapas (*n* = 3, 1.4%); Tabasco (*n* = 2, 0.9%); Veracruz (*n* = 1, 0.5%); ^‡^ Farmer, bricklayer, craftsman, carpenter, fisherman, painter, mechanic, cleaner, and domestic worker.

**Table 2 medicina-61-02032-t002:** The general characteristics of patients with gastric cancer during hospitalization.

Characteristics *n* (%)	Total *n* = 209	Men *n* = 115	Women *n* = 94	*p*-Value
Medical condition				
Wasting syndrome	140 (66.9)	73 (34.9)	67 (32.0)	0.223
Acid peptic disease	147 (70.3)	77 (36.8)	70 (33.5)	0.237
Bleeding from the upper digestive tract	18 (8.6)	7 (3.3)	11 (5.3)	0.712
Others ^†^	8 (3.8)	3 (1.4)	5 (2.4)	0.453
*Helicobacter pylori*	36 (17.2)	14 (6.7)	22 (10.5)	0.032
Blood type				
A Rh+	27 (12.9)	17 (8.1)	10 (4.8)	0.374
A Rh−	8 (3.8)	4 (1.9)	4 (1.9)	0.771
O Rh+	139 (66.5)	73 (34.9)	66 (31.6)	0.305
O Rh−	20 (9.6)	14 (6.7)	6 (2.9)	0.157
Others ^‡^	15 (7.1)	7 (3.3)	8 (3.8)	0.499
Anemia	84 (40.1)	50 (23.9)	34 (16.2)	0.284
Hypertension	31 (14.8)	10 (4.8)	21 (10.0)	0.006
Diabetes Mellitus	31 (14.8)	16 (7.6)	15 (7.2)	0.679
Dyslipidemia	22 (10.5)	12 (5.7)	10 (4.8)	0.340
BMI classification				
Underweight	31 (14.8)	16 (7.6)	15 (7.2)	0.679
Normal Weight	108 (51.6)	60 (28.7)	48 (22.9)	0.873
Overweight	54 (25.8)	32 (15.3)	22 (10.5)	0.468
Obese	16 (7.6)	7 (3.3)	9 (4.3)	0.346
Type of cancer				
Adenocarcinoma	203 (97.2)	112 (53.7)	91 (43.5)	0.802
Others ^§^	6 (2.8)	3 (1.4)	3 (1.4)	0.802

^†^ Dyspnea (*n* = 3, 2.8%); bleeding from the lower digestive tract (*n* = 2, 0.9%); ovarian tumor (*n* = 2, 0.9%); diarrhea (*n* = 1, 0.5%); ^‡^ AB Rh+ (*n* = 6, 2.9%); B Rh+ (*n* = 5, 2.4%); B Rh− (*n* = 4, 1.9%); ^§^ Lymphoma (*n* = 2, 0.9%); neuroendocrine tumor (*n* = 2, 0.9%); gastrointestinal stroma tumor (*n* = 2, 0.9%).

**Table 3 medicina-61-02032-t003:** The characteristics of the population according to endoscopic classification.

Characteristics *n* (%)	Borrmann Classification		*p*-Value
	(I–II) *n* = 28 (100)	(III–IV) *n* = 181 (100)	
Previous diagnosis of GC	11 (39.3)	64 (35.3)	0.687
Lifestyle (substance use behaviors)			
Tobacco uses	7 (25.0)	69 (38.1)	0.176
Alcohol consumption	15 (53.6)	91 (50.3)	0.746
Marijuana consumption	4 (14.3)	20 (11.0)	0.617
Medical condition			
Acid peptic disease	17 (60.7)	125 (69.1)	0.231
Wasting syndrome	16 (57.1)	124 (68.5)	0.234
Bleeding from the upper digestive tract	4 (14.3)	14 (7.7)	0.250
Others ^†^	1 (3.6)	7 (3.8)	0.423
Anemia	7 (25.0)	77 (42.5)	0.078
*Helicobacter pylori*	7 (25.0)	26 (14.4)	0.040
Consumption of proton pump inhibitors	16 (57.1)	133 (73.5)	0.156
BMI classification			
Underweight	4 (14.3)	27 (14.9)	0.097
Normal Weight	11 (39.3)	97 (53.6)	
Overweight	9 (32.1)	45 (24.9)	
Obese	4 (14.3)	12 (6.6)	
Stage of disease			
Localized tumor	16 (57.1)	41 (22.7)	0.001
Metastatic tumor	12 (42.9)	140 (77.3)	
Type of treatment received during hospitalization			
Medical treatment	11 (39.3)	78 (43.1)	0.705
Surgical treatment	17 (60.7)	103 (56.9)	

^†^ Borrmann I–II: Diarrhea (*n* = 1, 0.5%); Borrmann III–IV: Dyspnea (*n* = 3, 1.4%); bleeding from the lower digestive tract (*n* = 2, 0.9%); ovarian tumor (*n* = 2, 0.9%).

## Data Availability

Data underlying this work is available upon reasonable request. Request for data should be addressed to the corresponding author.

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
