# Peer review of "Clinicopathological Profiles and Survival Outcomes of Patients with Gastric Cancer According to the Borrmann Endoscopic Classification: A Single-Center Retrospective Cohort Study"

_medicina, 2025, doi:10.3390/medicina61112032_

Round 1

Reviewer 1 Report

Comments and Suggestions for Authors

I was glad to review this retrospective cohort study which investigates the clinical utility of the Borrmann classification in assessing survival outcomes in patients diagnosed with gastric cancer and treated at a specialized hospital in Mexico. The authors of this study concluded that the Borrmann classification remains a valuable and practical tool for evaluating gastric cancer. The manuscript is well-written and the topic is interesting. This article can be accepted for publication pending some minor corrections:

1) I would suggest adding extra information in the title of this manuscript ( prospective or retrospective study, single - center or multicentric). The title could be "Clinical Utility of the Borrmann Endoscopic Classification for Assessing Survival Outcomes in Patients with Gastric Cancer : A single-center retrospective cohort study"

2) In the abstract section, please add statistical information such as percentages, p-values in the results. In addition, you could add information on study population characteristics ( number of males , females, age, etc)

3) In Figure 1, I would suggest adding the specific reasons of exclusion

4) In Table 1, could you please add p-values?

5) Did you include patients with other malignant or benign tumor in this study (such as prostatic cancer)?

6) Explain the abbrevation EGJ in Figure 2

Author Response

Dear Reviewer, thank you for taking the time to review this manuscript and for providing your comments and suggestions. Please find below the answers to your comments and suggestions, point by point.

I was glad to review this retrospective cohort study which investigates the clinical utility of the Borrmann classification in assessing survival outcomes in patients diagnosed with gastric cancer and treated at a specialized hospital in Mexico. The authors of this study concluded that the Borrmann classification remains a valuable and practical tool for evaluating gastric cancer. The manuscript is well-written and the topic is interesting.

This article can be accepted for publication pending some minor corrections:

Comment 1: I would suggest adding extra information in the title of this manuscript (prospective or retrospective study, single - center or multicentric). The title could be "Clinical Utility of the Borrmann Endoscopic Classification for Assessing Survival Outcomes in Patients with Gastric Cancer: A single-center retrospective cohort study"

Response 1: We have incorporated your suggestion into the title. The new title is: “Clinical Utility of the Borrmann Endoscopic Classification for Assessing Survival Outcomes in Patients with Gastric Cancer: A single-center retrospective cohort study”

Comment 2: In the abstract section, please add statistical information such as percentages, p-values in the results. In addition, you could add information on study population characteristics (number of males, females, age, etc).

Response 2 : We have added the corresponding percentages, p-values, and relevant information on the study population characteristics (number of males and females, and age) to the Abstract.

Comment 3: In Figure 1, I would suggest adding the specific reasons of exclusion

Response 3: We have revised Figure 1 to include the specific reasons for patient exclusion in the study.

Comment 4: In Table 1, could you please add p-values?

Response 4: Thank you for your suggestion. We have added the p-value in Table 1

Comment 5: Did you include patients with other malignant or benign tumor in this study (such as prostatic cancer)?

Response 5: Thank you for your question. In this study, only patients diagnosed with gastric cancer were included. In addition, we have specified this point in the exclusion criteria described in the Materials and Methods section.

Comment 6: Explain the abbrevation EGJ in Figure 2

Response 6: Thank you for your suggestion. We have revised Figure 2 and replaced the abbreviation “EGJ” with “Esophagogastric junction”.

Reviewer 2 Report

Comments and Suggestions for Authors

The manuscript titled “Clinical Utility of the Borrmann Endoscopic Classification for Assessing Survival Outcomes in Patients with Gastric Cancer” is well written and presented.

This study shows that Gastric cancer (GC) is highly prevalent in southeastern Mexico and Borrmann classification continues to serve as a reliable, practical method for assessing GC. The median overall survival of 18 months highlights the late stage at diagnosis and the consequent high mortality associated with the disease.

This manuscript is recommended for acceptance after a minor revision to improve clarity and incorporate specific feedback, authors please provide explanations:

  1. There is no control in this study over confounding variables such as treatment differences, socioeconomic factors, or comorbidities.
  2. The study focuses on patients from a single specialized hospital in the Yucatán Peninsula. This may not represent the broader population of southeastern Mexico or other regions.
  3. The analysis does not mention multivariate adjustments (e.g., Cox regression) to control for age, sex, treatment type, or comorbidities when assessing survival by Borrmann type.
  4. It is unclear whether all patients received similar management (surgery, chemotherapy, or palliative care), which could strongly affect survival outcomes.
  5. The study is limited to Borrmann classification, a macroscopic description. It omits molecular subtypes, histological grades, or biomarkers (e.g., HER2, MSI, EBV), which are now clinically significant in GC prognosis.
  6. Only patients who underwent endoscopy and were confirmed histologically are included. This could exclude patients with non-endoscopically diagnosed or advanced inoperable disease.

Author Response

Dear Reviewer, thank you for taking the time to review this manuscript and for providing your comments and suggestions. Please find below the answers to your comments and suggestions, point by point.

The manuscript titled “Clinical Utility of the Borrmann Endoscopic Classification for Assessing Survival Outcomes in Patients with Gastric Cancer” is well written and presented.

This study shows that Gastric cancer (GC) is highly prevalent in southeastern Mexico and Borrmann classification continues to serve as a reliable, practical method for assessing GC. The median overall survival of 18 months highlights the late stage at diagnosis and the consequent high mortality associated with the disease.

This manuscript is recommended for acceptance after a minor revision to improve clarity and incorporate specific feedback, authors please provide explanations:

Comment 1: There is no control in this study over confounding variables such as treatment differences, socioeconomic factors, or comorbidities.

Response 1: Thank you for your comment. In response, we have reorganized the “Selection of Study Participants” section in the Materials and Methods, to clearly specify the study variables and minimize potential confusion in the manuscript.

Comment 2: The study focuses on patients from a single specialized hospital in the Yucatán Peninsula. This may not represent the broader population of southeastern Mexico or other regions.

Response 2: The hospital host is the only specialized center that receives patients without social security coverage from all three states of the Yucatan Peninsula (southeastern Mexico). We have added this explanation to the Discussion section: “The study was conducted at a specialized hospital in southeastern Mexico that is part of the Mexican government’s health services. The hospital’s area of influence covers the states of Yucatán, Campeche, and Quinata Roo, with a total population of more than 3 million inhabitants, making it a referral hospital in the region [17]”.

Comment 3: The analysis does not mention multivariate adjustments (e.g., Cox regression) to control for age, sex, treatment type, or comorbidities when assessing survival by Borrmann type.

Response 3: We have added the Cox regression analysis to assess survival, adjusted by sex, age, treatment type, and comorbidities. The corresponding results have been included in Table 4 of the Results section.

Comment 4: It is unclear whether all patients received similar management (surgery, chemotherapy, or palliative care), which could strongly affect survival outcomes.

Response 4: Thank you for your observation. To provide clarity and avoid potential confusion, we have specified the disease stages and the types of treatment received during hospitalization in Table 3. Additionally, we have detailed the supplementary treatments received by all patients, regardless of their primary treatment (chemotherapy, chemoradiotherapy, or palliative care).

Comment 5: The study is limited to Borrmann classification, a macroscopic description. It omits molecular subtypes, histological grades, or biomarkers (e.g., HER2, MSI, EBV), which are now clinically significant in GC prognosis.

Response 5: We appreciate the reviewer’s comment regarding molecular subtypes, histological grades, and biomarkers such as HER2, MSI, or EBV. While these factors are indeed clinically significant for gastric cancer prognosis, evaluating them was beyond the scope of our current study, which focused specifically on the clinical utility of the Borrmann endoscopic classification.

Comment 6: Only patients who underwent endoscopy and were confirmed histologically are included. This could exclude patients with non-endoscopically diagnosed or advanced inoperable disease.

Response 6:  We have added the exclusion criteria to the “Selection of Study Participants” section in Materials and Methods. In addition, Figure 1 has been updated to include specific exclusions, such as patients with non-endoscopic diagnoses or missing variables.